# Diagnostic yield of exome sequencing in nonobstructive azoospermia (NOA): A systematic review and meta-analysis

Fan Zhou[1,2], Yaqian Li[2,3], Jiani Zhang[2,3], Xiaodong Wang[2,3]*

1 Department of Medical Genetics/Prenatal Diagnostic Center, West China Second University Hospital, Sichuan University, Chengdu, Sichuan, China, 2 Ministry of Education, Key Laboratory of Birth Defects and Related Diseases of Women and Children (Sichuan University), Chengdu, Sichuan, China 3 Department of Obstetrics and Gynecology, West China Second University Hospital, Sichuan University, Chengdu, Sichuan, China

* wangxd_scu@sina.com

## Abstract

Azoospermia is considered as the most severe form of male infertility. The application value of exome sequencing (ES) in males with non-obstructive azoospermia (NOA) remains unclear. This study aims to review the known genetic causes of NOA and evaluate the diagnostic yield of ES in males diagnosed with idiopathic NOA. We performed a systematic database search in Ovid MEDLINE, EMBASE, CINAHL, Scopus, Cochrane Central Register of Controlled Trials, and Web of Science from database inception to March 2025. Two independent reviewers assessed the literature and included those studies investigating the utility of ES testing in men diagnosed with NOA and fulfilling the eligibility criteria. The pooled diagnostic yield was calculated using single-proportion analysis with random–effects modeling, and confidence intervals (CI) were estimated using the Clopper–Pearson exact method. A total of nine studies were included, and the qualities were assessed to be moderate to high via the modified STARD. Among the cohorts analyzed (nine studies comprising 1,728 individuals with NOA), the overall diagnostic yield of ES testing was 15% (95% CI: 10%–20%; low-certainty evidence). Of the 270 positive cases identified through ES testing, mutations in 262 genes were detected, with *AR, TEX11, FANCM, TDRD9, PNLDC1, M1AP, FBXO15,* and *DMRT1* being the most frequently observed. Among these cases, only 11.11% (5/45) reported successful testicular sperm extraction. The considerable heterogeneity indicates that the pooled prevalence estimates of the diagnostic yield of ES testing in NOA—approximately 15%—may overestimate the true diagnostic rate in the general NOA population. This estimate should thus be interpreted as an average across diverse clinical and methodological contexts, rather than a precise point estimate reflecting a uniform underlying effect. Future research, particularly large-scale studies using standardized protocols, is crucial to generate more accurate, reliable, and generalizable estimates of the diagnostic yield of ES testing in NOA.

**Data availability statement:** All relevant data are within the manuscript and its Supporting Information files.

**Funding:** The author(s) received no specific funding for this work.

**Competing interests:** The authors have declared that no competing interests exist.

## Introduction

The World Health Organization (WHO) recognizes male infertility as a significant global health issue, affecting approximately 15% of couples worldwide [1]. Among the various causes of male infertility, azoospermia is considered the most severe form, defined as the complete absence of spermatozoa in the ejaculate following centrifugation and subsequent microscopic evaluation of the specimen on two separate semen analyses [2]. Azoospermia is anatomically classified into three categories: pre-testicular, testicular, and post-testicular etiologies. Both pre-testicular and testicular azoospermia fall under the classification of non-obstructive azoospermia (NOA), which results from a failure of sperm production (either no mature sperm or only a few) or incomplete spermatogenesis [2]. In contrast, obstructive azoospermia (OA) is primarily caused by congenital or acquired obstruction of the urogenital tract, preventing the release of sperm despite normal sperm production.

NOA due to testis failure accounts for approximately 49–93% of men with azoospermia. Histopathological evaluation of testis biopsies in NOA cases typically reveals one of the following three patterns: hypospermatogenesis, maturation arrest (MA), and Sertoli cell-only (SCO) syndrome [2–3]. Genetic testing plays a critical role in revealing the etiology of male infertility, particularly in cases of NOA. Well-established genetic tests for males with azoospermia include karyotype analysis and Y chromosome microdeletion testing. Common genetic abnormalities identified in azoospermic males include Klinefelter syndrome (47, XXY) and chromosomal translocations or inversions [4–5]. Y chromosome microdeletions, particularly in the AZFa, AZFb, and AZFc regions, are found in approximately 7.5% of azoospermic males [6]. However, even with routine karyotype analysis and Y chromosome microdeletion testing, the genetic diagnostic yield for NOA males is only around 20% [5], indicating that the overwhelming majority of NOA individuals remain genetically undiagnosed.

Defects at any stage of spermatogenesis can lead to NOA. In cases of congenital pre-testicular factors, NOA is primarily managed with gonadotropin therapy, which achieves spermatogenesis in approximately 90% of males [2]. However, when NOA results from testicular failure or when gonadotropin therapy proves ineffective, testicular sperm extraction (TESE) is widely considered the first-line treatment option for sperm retrieval, followed by in vitro fertilization (IVF), as recommended by the Canadian Urological Association guidelines [2]. Nevertheless, surgical sperm retrieval, being invasive, costly, and potentially psychologically distressing for patients, identifies rare sperm in only approximately 52% of cases [7]. Failed TESEs may impose significant physical, financial and emotional burdens on infertile couples. Although preliminary prediction of TESE success rates may be informed by factors such as medical history, physical examination, endocrine hormone levels, ultrasonography, karyotype analysis, and Y-chromosomal microdeletion testing, the overall success rate of TESE remains modest, at approximately 50% [8]. Importantly, this approach often fails to incorporate comprehensive causality assessment through genetic testing. For example, in cases of germ cell arrest caused by underlying genetic abnormalities—where no mature spermatozoa are produced—TESE followed by IVF is

inherently unviable. Therefore, identifying specific causal genetic alterations holds significant potential for improving the prediction of sperm retrieval success and assessing the feasibility of overcoming infertility via IVF [9–10].

Emerging evidence suggests that more comprehensive genetic analyses, such as exome sequencing (ES), can potentially improve diagnostic yields and provide prognostic assessment in idiopathic NOA. An exome-based panel targeting genes strongly linked to male infertility (including *AR*, *DMRT1*, *M1AP*, *TEX11*, and *NR5A1*) has demonstrated a diagnostic yield of 8.5% in previously unexplained cases of azoospermia [10]. Integrating ES into routine clinical practice can provide a more comprehensive understanding of the genetic basis of NOA and help guide personalized treatment strategies, including the likelihood of successful sperm retrieval and IVF outcomes. This systematic review aims to evaluate the performance of exome sequencing in men with unexplained NOA and to provide evidence supporting the clinical application of advanced genetic technologies in male infertility.

## Materials and methods

This study was conducted in accordance with the Preferred Reporting Items for Systematic Reviews and Meta-Analyses (PRISMA) 2020 guidelines [11]. The study protocol was registered in the International Prospective Register of Systematic Reviews (PROSPERO; CRD420250655930).

### Study selection and data extraction

Studies reporting ES testing in males with NOA were included. The following databases were searched for eligible studies: Ovid MEDLINE, EMBASE, CINAHL, Scopus, Cochrane Central Register of Controlled Trials, and Web of Science. Additionally, reference lists of included studies were reviewed to identify additional eligible studies. The final search was conducted in March 2025. Detailed search strategies are provided in S1 Text.

Two independent reviewers screened titles, abstracts, and assessed full-text articles for inclusion. Disagreements were resolved by discussion or consultation with a third reviewer. Abstracts of all retrieved articles were initially screened based on selection criteria. For studies meeting the inclusion criteria, full-text articles were thoroughly evaluated and included in the final analysis. Inclusion criteria are as follows: studies reporting ES testing in participants with NOA; NOA was diagnosed according to WHO criteria; all reported variants were classified according to clinical guidelines, such as the American College of Medical Genetics and Genomics (ACMG) classification [12]. Exclusion criteria include studies with incomplete data or overlap with other studies, family-based studies, or studies reporting mutations on designated genes related to NOA.

### Quality assessment

Included studies were assessed for quality using a modified Standards for Reporting of Diagnostic Accuracy (STARD) checklist. Two reviewers independently evaluated each study, with discrepancies resolved through discussion. The quality assessment items included: a) eligibility criteria, inclusion criteria well defined; b) source of patients described; c) genetic tests, karyotype analysis and Y chromosome microdeletion performed; d) genetic testing approach described; e) data analysis approach of genetic testing described; f) Sanger validation described; g) variants classified strictly according to clinical guidelines; h) clinical characteristics of participants described; i) diagnostic variants, including pathogenic (P) and likely pathogenic (LP), listed. Additionally, reporting biases were evaluated based on the listed monogenic variants.

### Data synthesis

Data were extracted into a pre-designed form by two authors and reviewed for accuracy. The primary outcome was the diagnostic yield, defined as the percentage of gene variants related to NOA and classified into P/LP, as reported by included studies. Extracted data were pooled in a meta-analysis. The pooled effect size was calculated using

single-proportion analysis with random–effect modeling. Confidence interval (CI) was calculated using the Clopper–Pearson exact method [13–14]. Incidence rates with corresponding 95% CI and prediction intervals were estimated. Heterogeneity among studies was assessed using Tau$^2$ and I$^2$ statistics, with logarithmic transformation used as a variance-stabilizing transformation for proportions if necessary. Publication bias was assessed using funnel plots and the linear regression asymmetry test. The certainty of evidence was assessed by GRADE Working Group grades of evidence. Subgroup analyses were performed to explore potential sources of heterogeneity. A sensitivity analysis was conducted by excluding each study to assess the impact of methodological quality differences. All statistical analyses were performed using R Studio v1.0.136 (The R Foundation for Statistical Computing; meta v4.2 package).

## Results

### Eligible studies

Of the 1,506 records identified in database searching, nine studies fulfilled our eligibility criteria of an evaluation of the diagnostic performance of ES testing on males with NOA [15–23]. Ten studies were excluded from the meta-analysis due to family-based reports [24–25], genome-wide transcripts analysis [26], incomplete data [27–31], gene burden association testing [32], or because obstructive azoospermia males were also included [9]. (Fig 1).

A total of nine studies, encompassing 1,728 males clinically diagnosed with idiopathic NOA, were included in the synthetic analysis. The key characteristics of the included studies were tabulated descriptively in Table 1. All included studies were published after 2020. Specifically, four studies were conducted in Asia (three in China and one in Japan) [15,18,21,23], three studies were conducted in Europe (one in France, one in Poland, and the other one in Estonia) [16,17,22], one study was conducted in North Africa [19], and the remaining study was conducted with NOA cases recruited from 11 centers across the world [20]. Sample sizes of included studies ranged from 13 to 924 males diagnosed with idiopathic NOA.

### Quality assessment

All included studies clearly defined the eligibility criteria and described the source of patients. All studies reported the eligibility criteria, described the source of patients, exome sequencing approach, and analysis approach. Two studies did not describe the Sanger validation of positive monogenic variants [17,21]. Genetic variants classification was reported across all included studies using current guidelines (such as ACMG classification and ClinGen criteria). Two studies did not describe the clinical characteristics of individuals with idiopathic NOA [7,23]. All studies provided a list of diagnostic variants. The quality of all included studies was assessed to be moderate to high via the modified STARD (Fig 2).

### The diagnostic yield evaluated in the meta-analysis

A total of 1,728 males clinically diagnosed with idiopathic NOA were included in this meta-analysis. The diagnostic yield of ES testing in individuals with NOA is 15% (95% CI: 10%–20%, I$^2$ = 90%, 1,728 participants, 9 studies, low-certainty evidence) (Fig 3, S2 Text). The linear regression asymmetry test showed no significant quantification of bias (bias = 1.43, $P$ = 0.46). Publication bias was empirically depicted by the funnel plot (S3 Fig). Of the 270 cases with positive results from ES testing, 262 genes are involved (S4 Table). The top eight genes are *AR* (n = 8), *TEX11* (n = 6), *FANCM* (n = 5), *TDRD9* (n = 5), *PNLDC1* (n = 3), *M1AP* (n = 3), *FBXO15* (n = 3), and *DMRT1* (n = 3).

Because the patient cohorts of two studies overlapped and one included study reported genes without a NOA pheno-type [20,22], which is substantially different from the other included studies. We excluded this study and reanalyzed the data [20], resulting in a decrease in heterogeneity to 79%. The diagnostic yield of ES testing in this subgroup was 14% (95% CI: 8%–19%, 804 participants, 8 studies, very low-certainty evidence) (Fig 3, S2 Text). A linear regression asymmetry test revealed a significant quantification of bias (bias = 2.95, $P$ = 0.01). Publication bias was empirically illustrated

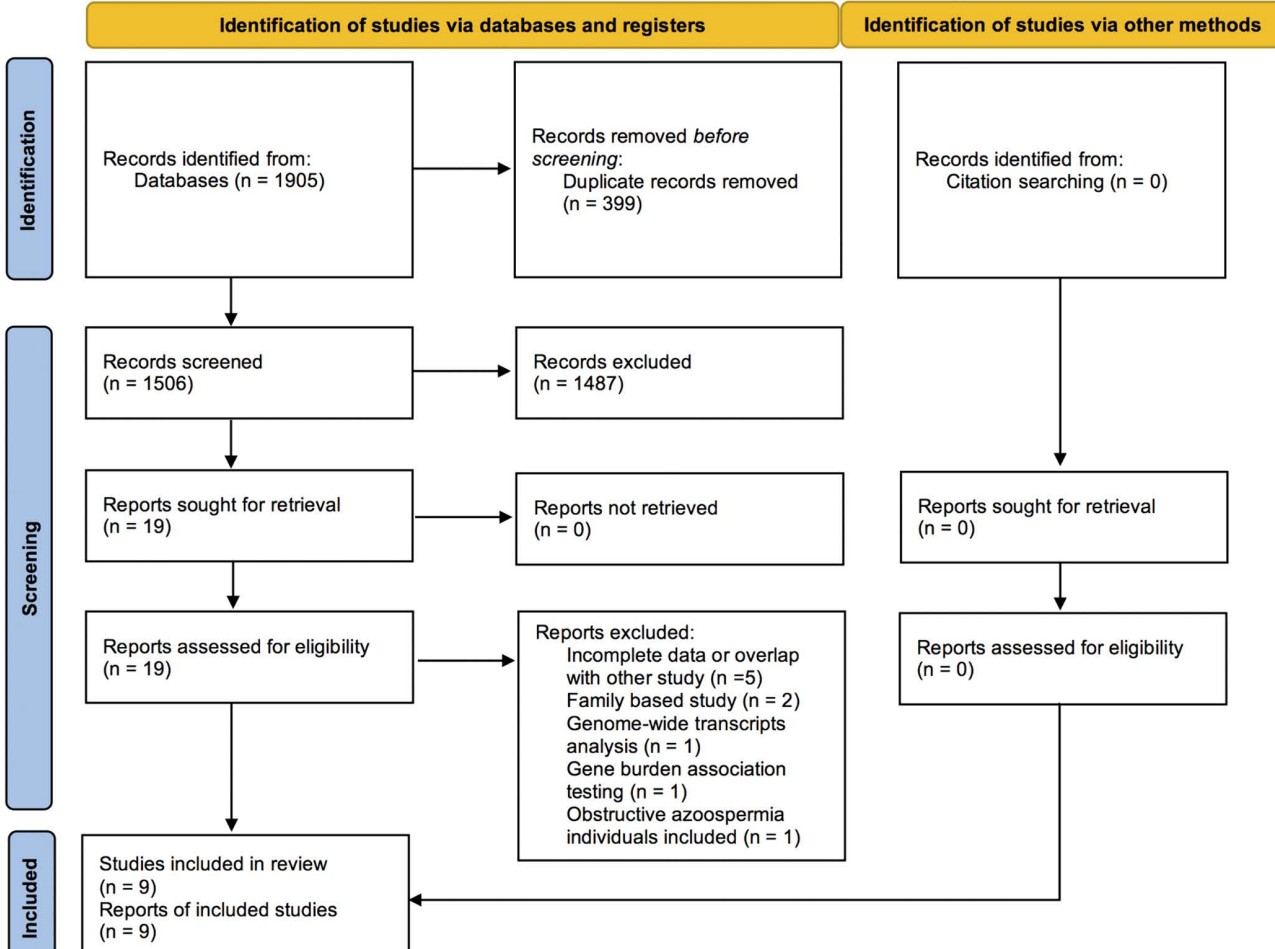

**Fig 1. PRISMA 2020 flow diagram for new systematic reviews, which included searches of databases, registers, and other sources.**

using the funnel plot (S3 Fig). The results of sensitivity analyses, generated by sequentially excluding each included study, along with the GRADE evidence profile for each pooled analysis assessing the diagnostic yield of ES in NOA, are provided in S2 Text.

Of the 270 cases with positive ES testing, 66 cases presented with MA (including MeA and poMeA), 62 cases presented with SCO, 12 cases presented with hypospermatogenesis, and 130 cases lacked histopathological examination. Additionally, TESE outcomes were reported for 45 cases, of which only 11.11% (5/45) had successful sperm retrieval (S4 Table).

## Gene involved in individuals with NOA reported in other literature

To investigate whether specific testicular histological phenotypes are associated with particular genetic variants, we systematically reviewed the genotype- phenotype correlations in males diagnosed with NOA. The implicated genes are summarized in Table 2. These genes were categorized according to their associations with specific testicular histopathological patterns underlying the NOA phenotype. Regarding sperm retrieval outcomes, favorable results were observed in

**Table 1. Characteristics of Studies Included in the Meta-analysis of ES testing for NOA.**

| Study | Participants | Genetic testing strategy | Variants classification | TESE | Diagnostic yield, No./total No. (%) |
|---|---|---|---|---|---|
| Chen 2020 [15] | 291 unrelated NOA patients of Chinese Han origin | Proband-only ES confirmed by Sanger sequencing | Yes | N/A | 14/291 |
| Ghieh 2022 [16] | 23 patients with idiopathic NOA in France after a TESE procedure* | ES validated with Sanger sequencing | N/A | Yes | 8/23 |
| Hardy 2022 [17] | 35 NOA individuals negative in Poland | ES was performed | Yes | N/A | 3/35 |
| Tang 2022 [18] | 55 unrelated idiopathic NOA patients in China | ES was performed | N/A | N/A | 8/55 |
| Kherraf 2022 [19] | 96 unrelated men originating from North Africa affected by non-syndromic NOA | ES was performed and bioinformatics analysis was limited to a panel of 151 genes selected as known causal or candidate genes for NOA | N/A | Yes | 22/96 |
| Nagirnaja 2022 [20] | 924 unrelated men diagnosed with unexplained NOA | ES was performed | Yes | Yes | 178/924 |
| Shi 2024 [21] | 13 NOA patients in China with a testicular volume > 6 mL | Underwent ES to investigate genetic factors | Yes | N/A | 3/13 |
| Lillepea 2024 [22] | 185 men with NOA of white European ancestry and living in Estonia. | ES data generation was performed and 638 candidate genes were analyzed | Yes | N/A | 20/185 |
| Muranishi 2024 [23] | 115 Japanese patients with isolated NOA and absence of sperm in microscopic-assisted TESE | ES was performed | Yes | Yes | 14/106 |

Note: * Seven consanguineous patients and in three of the 19 non-consanguineous patients.

NOA, nonobstructive azoospermia; ES, exome sequencing.

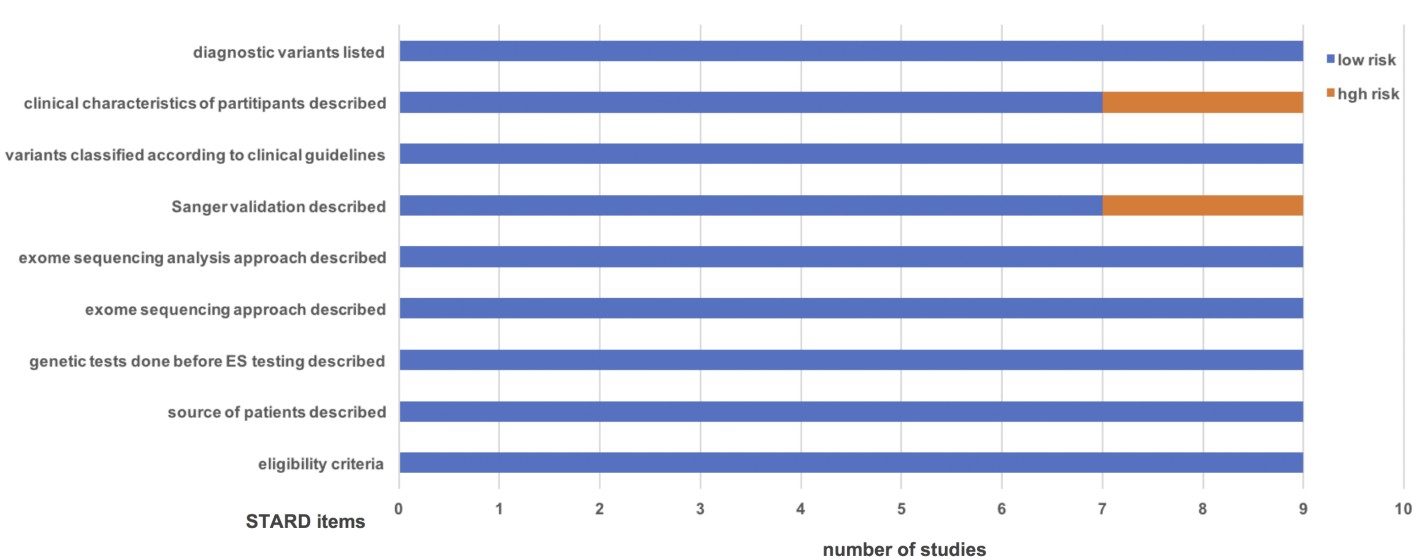

**Fig 2. Quality assessment of the nine studies included in this meta-analysis.** STARD, Standards for Reporting of Diagnostic Accuracy.

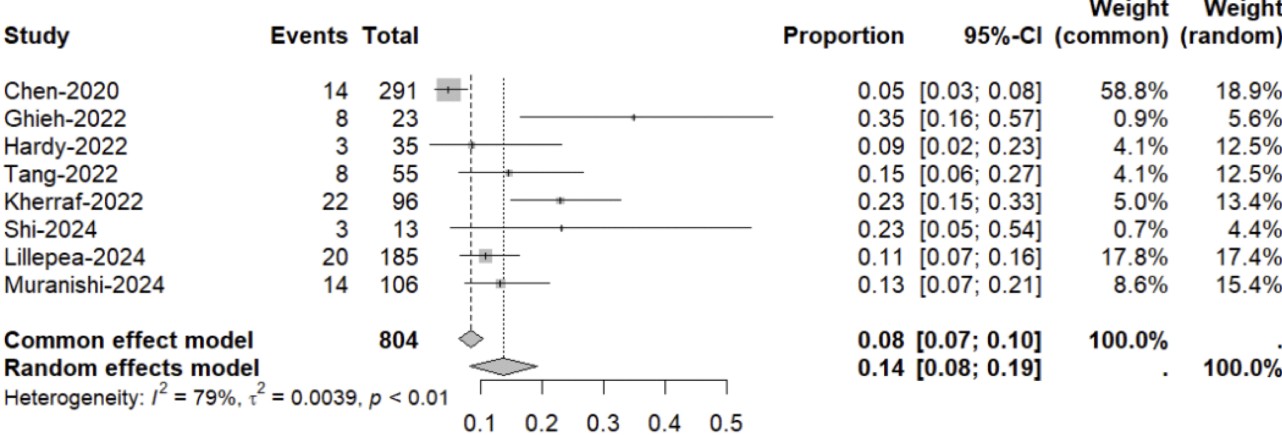

**Fig 3. Forest plot demonstrating the pooled diagnostic yield of ES testing in males with non-obstructive azoospermia (NOA).** Panel A, nine studies comprising 1,728 individuals; Panel B, eight studies comprising 804 individuals.

patients exhibiting hypospermatogenesis as well as those carrying variants in *HENMT1*, *PDHA2*, and *TDRD9*. In cases of maturation arrest, both successful and unsuccessful sperm retrievals have been reported among individuals with *DMC1* variants. In cases of Sertoli cell-only syndrome, positive sperm retrieval outcomes have been reported in individuals with *DMRT1* variants.

## Discussion

### Diagnostic yield and heterogeneity

This systematic review and meta-analysis evaluated the diagnostic yield of ES in idiopathic NOA males, and provided evidence on the clinical application of ES. Overall, nine studies with 1,728 participants were included. The results indicated that the diagnostic yield of ES in individuals diagnosed with NOA is 15% (95% CI: 10%–20%), and after excluding

**Table 2. Genes involved in NOA with known histopathological classification.**

| Histopathological classification | Literatures | Involved genes |
|---|---|---|
| Hypospermatogenesis | Studies included in this meta-analysis | ASZ1, *HENMT1*, *PDHA2*, TEX11, *TDRD9* |
| | Other studies | – |
| Maturation arrest | Studies included in this meta-analysis | **AR**, **ARL2**, ASZ1, **C11orf80**, **C14orf39**, **CCDC36**, CEP290, CHD5, **CHD7**, **CTCFL**, DCAF12L1, **DDX25**, DDX3Y, DHX37, DHX38, **DMC1**, **DMRT1**, ESX1, FANCA, **FANCM**, GATA4, **GCNA**, HAUS7, **HENMT1**, **HFM1**, **KISS1R**, M1AP, MAGEE2, **MCM8**, MCM9, **MCMDC2**, **MEI1**, **MEIOB**, MLH3, **MSH5**, MYRF, OTX2, **PDHA2**, PKD1, **PLK4**, PROKR2, **PSMC3IP**, **REC8**, SEMA3A, SMCHD1, **SYCE1**, **SYCP2**, **SYCP3**, **TDRD9**, **TDRKH**, **TERB1**, **TEX11**, TEX14, TGIF2LY, USP26, WNK3, WT1, ZFX |
| | Other studies | CCT6B [33], KASH5 [34], KCTD19 [35], MAJIN [36], MSH4 [37], NPAS2 [38], NR5A1 [39], PNLDC1 [40], RNF212 [41], SHOC1 [42], SOHLH1 [43], STAG3 [44], STX2 [45], TERB2 [36], TEX15 [46], XRCC2 [47] |
| Sertoli cell-only | Studies included in this meta-analysis | DHX37, DHX38, *DMRT1*, FANCA, FANCM, M1AP, MCM8, MYRF, OTX2, PLK4, PKD1, SMCHD1, TEX11, TEX14 |
| | Other studies | FSIP2 [48], NR5A1 [39], TEKT3 [49], USP26 [50] |

Note: Sperm retrieval results were marked as follows: positive results in italic bold, negative results in bold, and unknown results in non-bold. NOA, nonobstructive azoospermia.

one study with significant different reported criteria of detected variants, the diagnostic yield remained considerable at 14% (95% CI: 8%–19%). The high heterogeneity, with $I^2$ values ranging from 66% to 91% across analyses, suggests that clinical and methodological heterogeneity exists across the included studies. First, clinical heterogeneity may stem from variability in the characteristics of enrolled men with NOA. Although family-based studies were excluded from this meta-analysis, the remaining studies differed in key demographic and clinical parameters. Second, methodological heterogeneity—particularly in ES platforms and variant interpretation pipelines—represents a substantial source of variability. Despite performing a sensitivity analysis by sequentially excluding each study, significant heterogeneity persisted, indicating the presence of unmeasured or residual confounding factors. Furthermore, the observed funnel plot asymmetry, supported by a statistically significant Egger's regression test, suggests potential publication bias in one of the analyses. One plausible explanation is the "small-study effect", whereby smaller studies are often conducted in specialized centers with selective enrollment of patients, potentially leading to an inflated diagnostic yield. The considerable heterogeneity indicates that the pooled prevalence estimates of the diagnostic yield of ES testing in NOA—approximately 15%—derived from this meta-analysis may overestimate the true diagnostic rate in the general NOA population. This estimate should therefore be interpreted as an average across diverse clinical and methodological settings, rather than a precise point estimate reflecting a uniform underlying effect. Consequently, results should be interpreted with caution due to the possibility of upward bias. Future research, particularly large-scale studies employing standardized protocols, is essential to produce more accurate, reliable, and generalizable estimates of the diagnostic yield of ES testing in NOA.

## Genetic insights and gene-specific findings

NOA presented with high genetic heterogeneity in this meta-analysis, involving 262 genes across 270 positive cases. The most frequently mutated genes include *AR*, *TEX11*, *FANCM*, and *TDRD9*. Although the majority of these genes have been linked to male infertility or documented in the Mouse Genome Informatics (MGI) database, their association with NOA in humans and their clinical validity—assessed using the ClinGen Gene-Disease Validity framework—remain poorly established for most. Among the 262 reported genes, only 32 were classified as definitive, two as strong, one as moderate, four as limited, and one as refuted according to the ClinGen Gene-Disease Validity framework; the remaining 222 genes were not evaluated in the database. In the 270 cases with positive findings, pathogenic or likely pathogenic variants were

identified in 69 cases (25.56%), deleterious or loss-of-function (LoF) variants in an additional 29 cases (10.74%), and variants of uncertain significance (VUS) in 24 cases (8.89%). The remaining cases carried benign variants or those without definitive classification. No consistent patterns of increased recurrence rates or specific gene associations were observed across independent cohorts or subgroups.

Human spermatogenesis is a spatially intricate and asynchronous process, posing substantial challenges in the identification of multiple abnormalities during spermatid development and release. The human proteome map reveals elevated testis-specific expression of approximately 2,274 genes, among which 474 are exclusively detected in the testis. It has been estimated that around 625 monogenic causes of NOA may exist [20,51,52]. To date, over 200 genes have been associated with the NOA phenotype, with the majority implicated in meiotic processes—such as *TEX11*, *MEI1*, *DMC1*, and *EXO1* [8,16,19,20]. A smaller subset participates in other functional pathways, including DNA repair (e.g., *FANCM*), post-meiotic events (e.g., *DDX25*, *PDHA2*), and piRNA biogenesis (e.g., *TDRD9*) [8,53–55]. Among the most frequently mutated genes in this meta-analysis, androgen receptor (*AR*) gene mutations are known to cause androgen insensitivity syndrome. The AR plays a critical role in regulating kinase-mediated signaling pathways that control germ cell adhesion to Sertoli cells and the subsequent release of mature spermatozoa [56]. Mutations in *TEX11* impair the assembly and function of the synaptonemal complex, leading to severe defects in chromosome synapsis during the pachytene stage and dysregulation of the anaphase spindle checkpoint; these abnormalities lead to meiotic arrest, apoptosis of spermatocytes, and ultimately result in azoospermia [57]. *FANCM* is essential for genome replication and DNA repair under various conditions and participates in the ATR-mediated DNA damage response pathway [58]. *TDRD9* functions in the piRNA biogenesis pathway [20].

## Clinical correlation with TESE outcomes

NOA is typically diagnosed through semen analysis and ultrasound examination, and is clinically characterized by elevated follicle-stimulating hormone (FSH) levels and reduced testicular volume. Histologically, the condition is characterized by a substantially decreased quantity of germ cells, complete absence of germ cells in the testis, or arrest of spermatogenesis at specific stages of germ cell differentiation in testicular biopsies. Testicular biopsy with pathological classification has been proposed as a potential predictor of sperm retrieval success; however, it cannot be performed prior to invasive procedures, and individual patients may exhibit multiple spermatogenic defects or heterogeneous histological phenotypes [59]. Notably, TESE outcomes may differ from those predicted solely by diagnostic biopsy: sperm may be retrieved in some cases where only SCO syndrome is observed on biopsy, whereas no sperm may be found despite evidence of complete spermatogenesis in other cases [59–60]. Without a precise pathological or genetic diagnosis, no relatively reliable predictive information on the success rate of TESE could be provided to andrologists and affected males when facing the alternative choice of sperm donation. Genetic testing via ES offers a more convenient, feasible, and non-invasive approach for diagnosing, classifying, and predicting the likelihood of successful sperm retrieval in NOA. In this meta-analysis, patients with positive ES findings had a TESE success rate of only 11.11%, dramatically lower than the previously reported average of 27.10% in the literature [19]. However, the applicability of these findings may be constrained by the limited sample size and potential selection bias. Therefore, these findings should be interpreted as exploratory and hypothesis-generating rather than definitive. Larger prospective population-based studies are required to validate this association and accurately estimate its effect magnitude across diverse populations.

The success rates of TESE in men with azoospermia vary significantly depending on the underlying etiology and specific genetic factors involved. Patients with obstructive azoospermia (OA) generally exhibit high sperm retrieval rates with TESE followed by IVF, whereas those with NOA show markedly lower outcomes. Genetic variants affecting distinct stages of spermatogenesis—including meiosis, post-meiotic maturation, Piwi-related pathways, and DNA repair mechanisms—differentially impact the likelihood of successful sperm retrieval. In particular, sperm retrieval rates are comparatively low in individuals exhibiting maturation arrest. Notably, individuals carrying pathogenic

mutations in genes essential for meiosis are theoretically unlikely to yield viable sperm through TESE. Previous studies have shown that the histopathological pattern of hypospermatogenesis is associated with a nearly 100% TESE success rate, followed by 36.4% in testicular degeneration, 33.3% in post-meiotic arrest, and 16.7% in pre-meiotic arrest. The lowest success rates are observed in SCO syndrome (8.3%) and meiotic arrest (8.7%) [19]. Furthermore, mutations in *TEX11* gene account for approximately 1% of NOA cases, with variable histopathological phenotypes reported across cohorts; however, negative TESE outcomes are predominantly observed in individuals exhibiting maturation arrest [15,16,18,23,61]. The correlation between histopathological phenotypes and genetic etiology in individuals with NOA remains an active area of investigation.

### Broader implications and future directions

As of 2021, more than 120 genes have been identified as causative factors for male infertility, and this number continues to increase with accumulating evidence supporting the clinical validity of monogenic disorders associated with NOA [9]. Emerging evidence suggests that genetic testing can offer valuable prognostic insights and guide clinical decision-making regarding treatment options such as TESE or endocrine therapy. Therefore, a comprehensive genetic testing strategy is crucial for men with azoospermia to distinguish between OA and NOA (pre-testicular or testicular forms), establish a definitive genetic diagnosis, guide appropriate therapeutic interventions, and predict the likelihood of successful sperm retrieval—thereby avoiding unnecessary surgical procedures in cases with minimal or no chance of success. Furthermore, additional research is needed to strengthen genotype-phenotype correlations in the context of NOA-related male infertility.

### Strengths and limitations

This study has several strengths. Notably, it is the first meta-analysis to systematically investigate the diagnostic yield of ES testing in individuals with NOA. By synthesizing data from multiple studies that applied ES testing to males with NOA, the analysis encompassed a substantially enlarged sample size of 1,728 participants, thereby enhancing statistical power and generalizability. Nevertheless, certain limitations should be acknowledged. There was considerable heterogeneity across the included studies in terms of data analysis strategies—for instance, differences in whether analyses were restricted to specific gene panels—and in the criteria used to classify variants as clinically diagnostic. Furthermore, some studies reported genetic variants lacking well-established associations with NOA-related phenotypes. This meta-analysis identified significant heterogeneity in methodological design across studies. Upon excluding a potentially influential outlier study, statistical heterogeneity decreased to 79%, suggesting partial sensitivity to individual study effects. Therefore, further large-scale and rigorously designed studies are needed to standardize analytical protocols and generate more robust and consistent evidence.

## Conclusion

This meta-analysis helps to elucidate the contribution of monogenic variants in NOA by synthesizing data from studies that applied ES testing in individuals diagnosed with NOA. By identifying novel genes associated with human NOA, ES testing is promising in enhancing the precision of genetic diagnosis and possibly enabling more accurate assessment of the likelihood of successful sperm retrieval based on an individual's genotype. From this perspective, both men with NOA and healthcare providers may potentially benefit from further genetic testing. The diagnostic yield in NOA males is expected to increase further as additional monogenic causes are discovered and genotype–phenotype correlations become more comprehensively characterized. Large-scale prospective studies incorporating histological subtypes, sperm retrieval outcomes, IVF results, and both short-term and long-term follow-up data are needed to strengthen the evidence base for the clinical utility of genetic testing in individuals with NOA.

## Supporting information

**S1 Text. Search histories.**
(DOCX)

**S2 Text. GRADE evidence profile.**
(DOCX)

**S3 Fig. Funnel plot.**
(TIF)

**S4 Table. Gene variants reported in included studies.**
(XLSX)

**S1 File. PRISMA 2020 checklist.**
(DOCX)

## Author contributions

**Conceptualization:** Fan Zhou, Xiaodong Wang.

**Data curation:** Fan Zhou, Yaqian Li, Jiani Zhang.

**Formal analysis:** Fan Zhou, Yaqian Li, Jiani Zhang.

**Supervision:** Xiaodong Wang.

**Writing – original draft:** Fan Zhou.

**Writing – review & editing:** Yaqian Li, Jiani Zhang, Xiaodong Wang.

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
