## [Decision Letter · Decision Letter 0]

3 Nov 2025

Dear Dr. Wang,

Thank you for submitting your manuscript to PLOS ONE. After careful consideration, we feel that it has merit but does not fully meet PLOS ONE’s publication criteria as it currently stands. Therefore, we invite you to submit a revised version of the manuscript that addresses the points raised during the review process.

We look forward to receiving your revised manuscript.

Kind regards,

Su-Ren Chen

Academic Editor

PLOS ONE

Additional Editor Comments:

I agree with the comments from two reviewers and please address their concerns point-by-point in your revised manuscription.

Reviewers' comments:

Reviewer's Responses to Questions

**Comments to the Author**

1. Is the manuscript technically sound, and do the data support the conclusions?

Reviewer #1: Yes

Reviewer #2: Yes

2. Has the statistical analysis been performed appropriately and rigorously?

Reviewer #1: Yes

Reviewer #2: Yes

3. Have the authors made all data underlying the findings in their manuscript fully available?

Reviewer #1: Yes

Reviewer #2: Yes

4. Is the manuscript presented in an intelligible fashion and written in standard English?

Reviewer #1: Yes

Reviewer #2: Yes

Reviewer #1: The research is very important, but need some modification to be more relevant to the aim of the study.

Reviewer #2: The authors have written an important and timely meta-analysis addressing the diagnostic yield of exome sequencing (ES) in nonobstructive azoospermia (NOA) with rigorous methodology. However, there are several areas that require clarification and improvement:

1. GRADE Assessment Not Reported: While the methods mention that the certainty of evidence was assessed using GRADE criteria, the results of this assessment are not shared anywhere in the manuscript. Please include a summary of the GRADE findings, either in the main text or as a supplementary table.

2. Sensitivity Analysis Missing: Sensitivity analysis is described in the methodology section, but no results from such an analysis are presented in the results or supplementary material. Kindly clarify whether this was performed and, if so, summarize the findings.

3. Limited Exploration of Predictive Value for TESE: The study rightly mentions the limited predictive value of ES for testicular sperm extraction (TESE) outcomes, but this point is not explored in sufficient depth. A more nuanced discussion on how specific gene variants or histopathological phenotypes correlate with TESE success would enhance the clinical relevance of the findings.

4. Heterogeneity Not Addressed in Discussion: The meta-analysis demonstrates substantial heterogeneity (I² = 90%), yet this is not adequately addressed in the discussion section. Please elaborate on potential sources of heterogeneity (e.g., study design, population differences, sequencing methods, variant interpretation criteria) and their impact on the robustness of your conclusions.

Addressing these points will significantly improve the clarity, rigor, and impact of your manuscript.

**Do you want your identity to be public for this peer review?** For information about this choice, including consent withdrawal, please see our Privacy Policy

Reviewer #1: No

Reviewer #2: No

---

## [Author Response · Author response to Decision Letter 1]

13 Nov 2025

Reviewer #1

Introduction

1. Answer: Thank you for your valuable suggestions. We have streamlined the descriptions of micro-TESE techniques and IVF-ICSI protocols to focus more clearly on the genetic narrative. Page 3-4, line 69-84.

2. Answer: Thank you for your valuable suggestions. We have modified the relevant information in the revised version. Page 3-4, line 69-84.

3. Answer: Thank you for your valuable suggestions. We have modified the relevant information in the revised version. Page 3-4, line 69-84.

4. Answer: Thank you for your valuable suggestions. We have added the relevant information in the revised version. Page 4, line 87-88.

Results

1. Answer: Thank you for your detailed suggestions. We have revised them in the updated version. Page 7, line 164, line 173.

2. Answer: Thank you for your detailed suggestions. We have revised them in the updated version. Page 8, line 179.

3. Answer: Thank you for your detailed suggestions. We have revised them in the updated version. Page 7, line 168.

Main Concern:

1. Answer: We appreciate your valuable suggestions and have addressed the potential sources of heterogeneity by providing a clearer discussion in the revised manuscript. Page 8-9, line 198-206.

Due to the limited number of included studies and small sample size of these studies, we did not perform subgroup analysis or meta-regression; we acknowledge this as a limitation.

2. Answer: We appreciate your detailed suggestions. In the revised manuscript, we have discussed the increased risk of publication bias following the exclusion of the study with the largest sample size. This may be attributed to the "small-study effect," whereby smaller studies are often conducted in specialized centers that selectively enroll patients with more severe or familial phenotypes, potentially leading to an inflated diagnostic yield. Page 8, line 206-209.

3. Answer: We gratefully acknowledge the reviewer's insightful suggestions and have fully incorporated the feedback by revising the discussion section in accordance with the recommendations. Page 9-10, line 222-229; Page 10, line 234-237.

Discussion

1. Answer: We gratefully acknowledge your valuable suggestions and have accordingly divided the discussion section as recommended. Page 8-12.

2. Answer: We appreciate your thoughtful inquiry and have incorporated the relevant information in the revised manuscript to enhance clarity and completeness. Page 10, line 231-234.

3. Answer: We gratefully acknowledge your insightful suggestions. In accordance with your recommendation, we have incorporated a detailed discussion of the biological functions, clinical implications, and mechanistic relevance of the "top genes" in the Discussion section. Page 10, line 237-246.

4. Answer: Thank you for your valuable suggestions. In accordance with your recommendation, we have reinterpreted the TESE success rate observed in this meta-analysis, which is based on a small sample size. Page 11, line 263-266.

5. Answer: We appreciate your valuable suggestion and have removed the description regarding the AI model to maintain overall coherence. Page 12.

Reviewer #2

1. Answer: We gratefully acknowledge your valuable suggestions. A summary of the GRADE findings has been included in Supplementary File 2. Page 7-8, line 174-176.

2. Answer: We gratefully acknowledge your insightful suggestions. The sensitivity analysis, conducted by sequentially excluding individual studies, has been summarized and presented in Supplementary File 2. Page 8, line 174-176.

3. Answer: We gratefully acknowledge your valuable suggestions. In response, we have incorporated a more comprehensive discussion on the associations between specific gene variants or histopathological phenotypes and TESE outcomes, informed by current evidence. We further emphasize that the relationship between histopathological characteristics and underlying genetic etiologies in individuals with NOA remains an active area of investigation. Page 11-12, line 268-282.

4. Answer: We gratefully acknowledge your insightful suggestions. In the revised manuscript, we have thoroughly examined the potential sources of heterogeneity and carefully evaluated their implications for the robustness of our conclusions. Page 8-9, line 198-216.

---

## [Decision Letter · Decision Letter 1]

30 Nov 2025

Diagnostic Yield of Exome Sequencing in Nonobstructive azoospermia (NOA): A Systematic Review and Meta-analysis

PONE-D-25-31306R1

Dear Dr. Wang,

We’re pleased to inform you that your manuscript has been judged scientifically suitable for publication and will be formally accepted for publication once it meets all outstanding technical requirements.

Kind regards,

Su-Ren Chen

Academic Editor

PLOS ONE

Additional Editor Comments (optional):

The two reviewers have no further concerns and this revision is recommended for acceptance.

Reviewers' comments:

Reviewer's Responses to Questions

**Comments to the Author**

Reviewer #1: All comments have been addressed

Reviewer #2: All comments have been addressed

2. Is the manuscript technically sound, and do the data support the conclusions?

Reviewer #1: Yes

Reviewer #2: Yes

3. Has the statistical analysis been performed appropriately and rigorously?

Reviewer #1: Yes

Reviewer #2: Yes

4. Have the authors made all data underlying the findings in their manuscript fully available?

Reviewer #1: Yes

Reviewer #2: Yes

5. Is the manuscript presented in an intelligible fashion and written in standard English?

Reviewer #1: Yes

Reviewer #2: Yes

Reviewer #1: Thank you for efforts.

Reviewer #2: The authors have carefully and comprehensively addressed all major and minor comments raised during the previous round of review. The manuscript has undergone substantial improvement in clarity, structure, and scientific rigor.

Specifically:

The GRADE assessment has now been appropriately included as Supplementary File 2.

The sensitivity analysis, previously missing, has been performed and summarized clearly.

The discussion of TESE predictive value has been expanded with relevant clinical and mechanistic context.

The sources of heterogeneity are now thoroughly examined, with an appropriate acknowledgment of methodological limitations given the limited number of included studies.

The discussion has been reorganized for improved readability, and the expanded explanation of the biological relevance of key genes strengthens the interpretability of the findings.

All issues related to clarity, background focus, and unnecessary content (e.g., the AI model reference) have been resolved.

Overall, the manuscript now meets PLOS ONE’s standards for scientific soundness, clarity, transparency of reporting, and methodological rigor. I have no further concerns.

I recommend acceptance.

**Do you want your identity to be public for this peer review?** For information about this choice, including consent withdrawal, please see our Privacy Policy

Reviewer #1: **Yes: ** Weam Aldiban

Reviewer #2: No

---

## [Editor Report · Acceptance letter]

PONE-D-25-31306R1

PLOS One

Dear Dr. Wang,

I'm pleased to inform you that your manuscript has been deemed suitable for publication in PLOS One. Congratulations! Your manuscript is now being handed over to our production team.

Kind regards,

on behalf of

Prof. Su-Ren Chen

Academic Editor

PLOS One